

# An anomaly detection model for multivariate time series with anomaly perception

Dong Wei, Wu Sun, Xiaofeng Zou, Dan Ma, Huarong Xu, Panfeng Chen, Chaoshu Yang, Mei Chen and Hui Li

State Key Laboratory of Public Big Data, College of Computer Science and Technology, Guizhou University, Guiyang, China

## ABSTRACT

Multivariate time series anomaly detection is a crucial data mining technique with a wide range of applications in areas such as IT applications. Currently, the majority of anomaly detection methods for time series data rely on unsupervised approaches due to the rarity of anomaly labels. However, in real-world scenarios, obtaining a limited number of anomaly labels is feasible and affordable. Effective usage of these labels can offer valuable insights into the temporal characteristics of anomalies and play a pivotal role in guiding anomaly detection efforts. To improve the performance of multivariate time series anomaly detection, we proposed a novel deep learning model named EDD (Encoder-Decoder-Discriminator) that leverages limited anomaly samples. The EDD model innovatively integrates a graph attention network with long short term memory (LSTM) to extract spatial and temporal features from multivariate time series data. This integrated approach enables the model to capture complex patterns and dependencies within the data. Additionally, the model skillfully maps series data into a latent space, utilizing a carefully crafted loss function to cluster normal data tightly in the latent space while dispersing abnormal data randomly. This innovative design results in distinct probability distributions for normal and abnormal data in the latent space, enabling precise identification of anomalous data. To evaluate the performance of our EDD model, we conducted extensive experimental validation across three diverse datasets. The results demonstrate the significant superiority of our model in multivariate time series anomaly detection. Specifically, the average F1-Score of our model outperformed the second-best method by 2.7% and 73.4% in both evaluation approaches, respectively, highlighting its superior detection capabilities. These findings validate the effectiveness of our proposed EDD model in leveraging limited anomaly samples for accurate and robust anomaly detection in multivariate time series data.

## INTRODUCTION

In real-world scenarios, multiple sensors or agents continuously collect data from various system components for real-time monitoring (*Kim et al., 2023*). These sensors will generate a large amount of interrelated time series data. For example, in IT systems, performance

Corresponding authors
Mei Chen, gychm@qq.com
Hui Li, cse.huili@gzu.edu.cn,
huili.gm@gmail.com

monitoring software simultaneously gathers metrics like CPU utilization, memory usage, and user visits, among others, and stores them in a time series database. These metrics often exhibit complex correlations. For instance, an increase in user visits will usually lead to an increase in CPU utilization. When the system is abnormal, this correlation may be broken. However, manually identifying anomalies from hundreds or thousands of performance metrics based on experience alone is challenging and often impractical. Therefore, employing machine learning algorithms to detect anomalies in the multivariate time series data collected by these systems has emerged as a significant research area (*Cook, Msrl & Fan, 2020*).

Normal data in anomaly detection often consists of samples that exhibit regular patterns, while anomalies are characterized as any substantial deviation from the normal samples (*Tian, Su & Yin, 2022*). The data in time series anomaly detection has two categories: univariate and multivariate. Univariate time series anomaly detection focuses on identifying anomalies in a singular metric. However, anomalies in a single metric may not always be indicative of systematic anomalies, making this strategy ineffective for finding systematic anomalies. Conversely, multivariate time series anomaly identification involves examining the association between variables, providing a more precise reflection of anomalies at a systemic level. In this work, we focus on the issue of detecting anomalies in multivariate time series data.

Prior research often prefers unsupervised implementation due to the difficulty and cost of obtaining anomalous examples (*Li & Jung, 2023*). Their approaches involve training the model with several normal samples and subsequently utilizing either the reconstruction error (*Malhotra et al., 2016*; *Li et al., 2019*; *Su et al., 2019*) or prediction error (*Hundman et al., 2018*; *Zong et al., 2018*) to detect anomalies. As a result of the model receiving substantial training exposure to normal data, anomaly detection in the data can be achieved through a comparison of the reconstruction or prediction error for the normal data to that of the unexposed anomalous data.

For instance, *Malhotra et al. (2016)* employed an encoder–decoder scheme based on long short term memory (LSTM) networks to reconstruct time series and detect anomalies through reconstruction errors. In order to capture the potential interdependencies among the variables, *Li et al. (2019)* incorporated the entire collection of variables and utilized a GAN framework with an LSTM model to represent the temporal correlation of the series. An unsupervised anomaly identification technique was proposed by *Hundman et al. (2018)*. It makes use of LSTM networks to forecast temporal data and then exploits prediction errors to identify anomalies. The DAGMM model was proposed by *Ruff et al. (2019)* and focuses on multivariate data anomaly detection without temporal dependencies. It seamlessly integrates the processes of density estimation and dimensionality reduction in a natural way. Because the two processes are independent, end-to-end joint training prevents the model from achieving a local optimum and prediction errors are also used to identify anomalies. Although some progress has been made in the field of anomaly detection in time series data, the limited number of anomaly samples utilized in prior research have not been fully utilized. These samples contain valuable information that could provide more precise guidance for the anomaly detection model.

Addressing the challenges posed by multivariate time series anomaly detection, this paper presents a novel anomaly detection model called EDD (Encoder-Decoder-Discriminator). The EDD model aims to effectively utilize limited abnormal sample information to enhance the model's ability to identify anomalies. It achieves this by leveraging the differences in probability distributions between normal and abnormal data in the latent space, addressing the issue that reconstruction and prediction models can become too robust and insensitive to minor anomalies. In summary, our work offers an innovative solution to anomaly detection in multivariate time series, enhancing reliability and precision. The salient contributions of this work are outlined as follows:

- Novel anomaly detection technique (EDD): We introduced a new anomaly detection technique, EDD, tailored for multivariate time series data. This technique excels at extracting diverse features, enabling a more precise distinction between normal and abnormal patterns.
- Effective utilization of limited anomalous data: Traditional methods often overlook the limited availability of anomalous data in multivariate time series. We addressed this gap by effectively incorporating this data into our model. This approach is crucial for instructing algorithms to accurately differentiate between normal and anomalous instances, significantly improving the model's performance.
- Rigorous evaluation: To assess the performance of our model, we conducted extensive comparison experiments using two distinct identification approaches. These experiments were conducted on both publicly available datasets and a proprietary dataset, ensuring a comprehensive evaluation. The results demonstrate that our approach surpasses state-of-the-art methods in identifying abnormalities for multivariate time series data.

# RELATED WORK

In data analysis, the detection of anomalies in time series data stands as a pivotal concern that has garnered significant attention from researchers. To address this challenge, numerous strategies have been proposed to efficiently pinpoint these irregularities. These methods can be categorized into two broad families based on their underlying methodology: classical machine learning algorithms and deep learning algorithms.

## Classic algorithms

Because of their simplicity and interpretability, classical machine learning algorithms have been favored by some studies in the past for anomaly detection tasks. For instance, *Ramaswamy, Rastogi & Shim (2000)* introduced a new method for identifying outliers using the distance from a point to its $k$-th nearest neighbor. Points were ranked based on this distance, and the top $n$ points were classified as outliers. *Shyu et al. (2003)* employed the principal component analysis method to extract key features from the data, resulting in low-dimensional projections. The reconstruction error resulting from this process was then utilized as the anomaly score. Furthermore, *Liu, Ting & Zhou (2008)* proposed a novel model-based approach to anomaly detection that shifts the focus from profiling normal instances to explicitly isolating anomalies. The innovative iForest

algorithm efficiently leverages sub-sampling, resulting in linear time complexity and minimal memory requirements. While these techniques offer ease of use and minimal processing complexity, their detection accuracy is somewhat limited due to the neglect of the intrinsic temporal properties inherent in time series data.

## Deep learning algorithms

In recent years, with the expansion of data size and complexity, classical machine learning algorithms have faced performance bottlenecks when dealing with large-scale and high-dimensional data. Deep learning is widely used in the field of multivariate time series data anomaly detection due to its excellent feature extraction capability and superiority in modeling high-dimensional sequence data. Some recurrent neural networks, such as RNN and LSTM, are widely used in time series data anomaly detection due to their natural sequence feature extraction capability. For example, *Su et al. (2019)* proposed a stochastic recurrent neural network that captures the normal patterns of multivariate time series by modeling the data distribution using stochastic latent variables. *Wei et al. (2023)* proposed a hybrid model based on LSTMs and self-encoders that solves the long-term dependency problem, which cannot be solved by shallow machine learning. Due to its efficient parallel computing capabilities and powerful context information processing abilities, *Tuli, Casale & Jennings (2022)* chose to build an anomaly detection model based on Transformer. This model can leverage the characteristics of Transformer to quickly and accurately identify abnormal observations in time series data. In addition, in order to explicitly capture the dependencies between variables, *Deng & Hooi (2021)* has incorporated a graph structure learning module to increase the spatial feature extraction capability of the model. To increase model performance, several studies (*Zhao et al., 2020*; *Xia et al., 2023*) have, of course, merged the first two types of approaches to extract temporal as well as spatial correlations of sequences. More recently, diffusion models (*Lin et al., 2023*) have attracted significant attention in the realm of artificial intelligence content generation, with some researchers applying them to anomaly detection in multivariate time series data. A noteworthy example is ImDiffusion (*Chen et al., 2023*), a novel framework rooted in the imputation diffusion model. It accurately captures the inherent dependencies and stochastic nature of MTS data, enabling precise and robust anomaly detection. However, a notable limitation of these techniques is that they do not fully leverage the limited available aberrant data, which could potentially further enhance the performance of multivariate time series anomaly detection.

## PROBLEM STATEMENT

In this article, the model primarily receives multivariate time series data collected over a specific time frame as input. Consider an entity or system equipped with $N$ sensors, each capturing data at regular intervals. The amassed data from all sensors during a designated time period can be expressed as $X \in R^{N \times L}$, where $L$ represents the length of each individual time series. Furthermore, $X_{i,:}$ signifies the complete time series recorded by the $i$th sensor or feature, while $X_{:,t}$ indicates the values of all features captured at time point t. For each

time series, a sliding time window of length $k$ is used to generate a fixed-length input, and $X_{:,t-k+1:t}$ represents the time window data divided at time $t$.

In general, the task of multivariate time series anomaly detection involves utilizing the observation value at time $t$ to assess whether an entity exhibits anomalous behavior. This can be formally expressed through Eq. (1), which outlines the detection mechanism based on the available data.

$$y_t = f(X_{:,t}) \tag{1}$$

where $y_t$ is a Boolean indicator, a value of 1 signifies that the entity is in an abnormal state at the $t$-th timestamp, whereas a value of 0 implies the entity is operating normally. This binary representation enables clear identification of anomalous events within the multivariate time series data.

Recognizing the crucial role of historical data in analyzing the current state of an entity, the utilization of such data spanning a specific duration can be leveraged to enhance the performance of anomaly detection. Consequently, Eq. (1) can be refined to Eq. (2), incorporating the historical context to improve detection accuracy.

$$y_t = f(X_{:,t-k+1:t}). \tag{2}$$

Therefore, we devise a deep learning model that seamlessly integrates the comprehensive feature set of an entity at time $t$ alongside contextual information, ultimately outputting a precise determination of whether the entity exhibits anomalous behavior at that specific timepoint.

## THE EDD MODEL

### Overview

The goal of the EDD model is to encode normal data into similar probability distributions within the latent space, while ensuring that the probability distribution of abnormal data stands out distinctly. This approach allows for the precise identification of abnormalities. The EDD model is comprised of three key components: the Encoder, the Decoder, and the Discriminator. Figure 1 illustrates the model's fundamental structure. The Encoder is responsible for meticulously extracting the features from the multivariate time series data and encoding the normal data into a designated distribution. To ensure that the Encoder captures the essence of the original data and its output distribution incorporates the primary information, the Decoder is employed. The Decoder reconstructs the original data using samples drawn from the Encoder's probability distribution. The Discriminator of EDD is proposed to discern the disparities between the distributions of normal and anomalous data in the latent space. Its guidance also assists the Encoder in determining which type of data (specifically, anomalous data) does not require encoding into the designated distribution. Each of the three modules is comprehensively explained in the following subsections.

### Encoder

The Encoder is responsible for extracting relevant features from time series data. In analogy to VAE (*Kingma & Welling, 2013*), the EDD model aims to map these extracted features

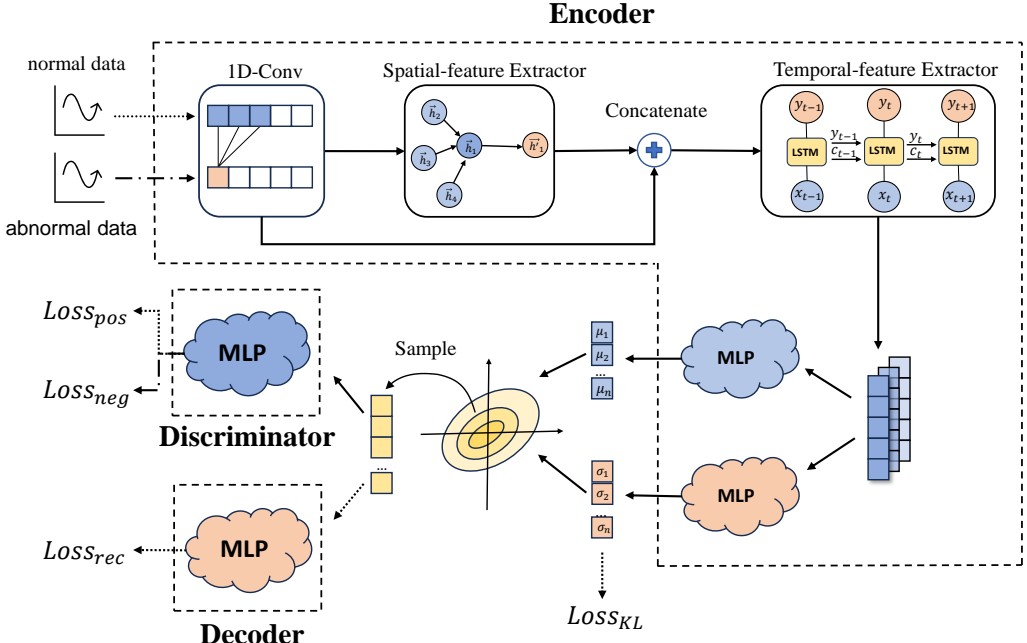

**Figure 1** EDD model architecture (arrows represent the transfer of data between model components, with normal data indicated by dotted line and abnormal data indicated by dash-dotted, and straight line represents both normal and abnormal data).

onto a Gaussian distribution. Over the long term, the dependencies among multivariate time series data remain consistent, with normal data adhering to these dependencies while abnormal data deviate from them. Consequently, theoretically, anomaly detection can be achieved by isolating the characteristics that are shared by normal data but absent in abnormal data. When extracting features from multivariate time series data, it is essential to capture both temporal and spatial features. Temporal features refer to the relationships between individual variables across different time points, elucidating the evolving trends within each sequence. Spatial features, on the other hand, highlight the interconnections among various variables. Accurate extraction of these two types of features enhances the model's generalization capabilities, enabling a more precise distinction between normal and abnormal data.

Prior research (*Dos Santos & Gatti, 2014*) has demonstrated the effectiveness of convolution operations in feature extraction. Therefore, we initiate the feature extraction process by applying a 1-D convolution layer to the original data. This step aids in preprocessing the data before proceeding to extract the aforementioned temporal and spatial features.

### Spatial-feature extractor

To extract associative relationships among variables, we utilize Graph Attention Networks (GAT) (*Veličković et al., 2017*). Initially, the data is represented as a graph structure, where

each variable corresponds to a graph node. The time series of variables within a given window serve as node features, with all nodes being neighbors of each other.

Subsequently, we employ GAT to learn the dependencies between these variables. GAT computes the similarity between nodes *via* an attention mechanism and assigns weights accordingly. Specifically, for each node, GAT computes similarity scores with its neighboring nodes, transforms these scores into probability distributions using the softmax function, and then weights and sums the neighboring node features based on the attention weights. These weighted features are subsequently combined with the node's own features to update its representation. This process can be formally expressed by the following formulas:

$$e_{ij} = LeakyReLU(\vec{a}^T[Wh_i||Wh_j]) \tag{3}$$

$$\alpha_{ij} = \frac{Softmax(e_{ij})}{\sum_{k \in N_i} Softmax(e_{ik})} \tag{4}$$

$$h_i' = \sigma\left(\sum_{j \in N_i} \alpha_{ij} Wh_j\right) \tag{5}$$

where $h_i$ denotes the feature vector of node $i$, $N_i$ denotes the set of neighboring nodes of node $i$, $W$ is the learnable weight matrix, $\vec{a}$ is the learnable attention vector, $\sigma$ is the activation function, and $h_i'$ is the feature representation with the dependency relationship between variables.

### Temporal-feature extractor

Recurrent neural networks (RNNs) possess remarkable feature extraction capabilities for sequential data. Among various RNN variants, the LSTM (*Hochreiter & Schmidhuber, 1997*) stands out as a powerful model that addresses the gradient disappearance issue in standard RNNs. The LSTM is capable of learning long-term dependencies within sequences, making it an ideal choice for extracting temporal dependencies. In this study, we utilize the LSTM network to capture the temporal dependencies in the given sequence. We integrate the output of the 1-D convolution layer with the spatial feature extraction module, resulting in a vector matrix $C$ with dimensions $k \times 2n$. This matrix $C$ is then fed into the LSTM model to extract the temporal dependencies. In essence, at each time step, the hidden state output $h_{t-1}$ of the previous unit, the memory state $c_{t-1}$, and the data of the current time step are jointly processed by the current LSTM unit, as shown in Eq. (6):

$$h_t, c_t = LSTM(e_t, h_{t-1}, c_{t-1}). \tag{6}$$

In our work, the hidden state output of the final LSTM unit encapsulates information from all time steps, and the hidden state output is forwarded to the latent distribution mapping module as spatiotemporal fusion features, enabling the model to effectively incorporate both spatial and temporal information for downstream tasks.

*Latent distribution mapper*

The Gaussian distribution is employed as a representative of the latent distribution of the data. The latent distribution mapping module is constructed from two identically structured multi-layer perceptrons (MLPs). These two MLPs are tasked with producing the mean and standard deviation of the Gaussian distribution in the latent space, respectively. This process can be expressed as follows:

$$Means = MLP_1(h_f) \tag{7}$$

$$Stds = MLP_2(h_f). \tag{8}$$

The symbol $h_f$ denotes the feature vector that integrates spatial and temporal information. To bolster the model's information-conveying capabilities, the multilayer perceptrons produce diverse sets of latent distributions as their output.

As previously mentioned, in order to distinguish between normal and abnormal data in the latent space, it is expected that the Encoder's outputs for normal data will be clustered at specific points, while the outputs for abnormal data will exhibit a random distribution. In order to achieve this, we employ a loss function to modify the output of the Encoder:

$$Loss_{KL} = KL(N(\mu, \sigma^2), N(\mu_0, \sigma_0^2)). \tag{9}$$

To ensure that the Encoder's processing of normal data adheres to the desired distribution pattern, we compute the KL divergence between the distribution of normal data outputs and a pre-specified Gaussian distribution. This KL divergence serves as one of the losses in the model. Subsequently, we employ the gradient descent algorithm to refine the model's parameters accordingly.

## Decoder

In our EDD model, a subset of data is randomly sampled from the Gaussian distribution formulated by the Encoder, leveraging the reparameterization strategy. These sampled data are subsequently fed into the Decoder, which serves as a multi-layer perceptron responsible for reconstructing the original data. The Decoder plays a pivotal role in guiding the Encoder towards extracting richer, more meaningful features. While the Decoder is not directly involved in anomaly detection, its significance lies in providing a mechanism that enables the Encoder to delve deeper into the inherent differences between normal and abnormal data in the training dataset. Without the Decoder, the Encoder may overrely on superficial features, disregarding the deeper, more significant features within the data. This limitation may compromise the model's generalization capabilities. Consequently, when confronted with unseen anomalies, the Discriminator may struggle to effectively identify them if it lacks adequate extraction of internal features.

We utilize a multi-layer perceptron (MLP) as the architecture for our Decoder. By introducing randomly sampled data into this MLP, it generates a reconstructed approximation of the original data. Subsequently, we employ gradient descent to minimize

the reconstruction loss, which is computed as the mean square error (MSE) between the reconstructed and the actual data. The reconstruction loss is formally defined as follows:

$$Loss_{rec} = \frac{\sum_{t=1}^{k} \sum_{i=1}^{N} (x_{i,t} - \hat{x}_{i,t})^2}{N \times k}. \tag{10}$$

In this formula, $x_{i,t}$ denotes the actual value of the $i$th variable at time $t$ within the considered time window, while $\hat{x}_{i,t}$ represents the reconstructed approximation of the same variable at the same time point.

## Discriminator

Just like the Decoder, the Discriminator obtains its input *via* reparameterization, which involves randomly sampling data from the Gaussian distribution generated by the Encoder. The core responsibility of the Discriminator is to accurately differentiate between normal and abnormal data by learning the distinct probability distributions of both types in the latent space. Additionally, since the gradients of the sampled data are not truncated, the Discriminator effectively guides the Encoder to refrain from encoding abnormal data into regions of high probability density. Unlike supervised learning methods that solely rely on a loss function to separate the Encoder's output for abnormal data, our approach offers the advantage of allowing the Encoder to focus solely on learning the features of normal data. This enhanced focus on normal data features boosts the model's expressive power, facilitating the clustering of normal data in the latent space. Consequently, the Discriminator can effectively discern the distributional disparities between normal and abnormal data in the latent space, even with limited exposure to abnormal data samples.

The Discriminator's architecture also employs a multi-layer perceptron (MLP) for implementation. The model's output comprises a set of vectors. To ensure precise anomaly detection, we aim for the model to produce vectors that are nearly all-zeros for normal data and close to all-ones for abnormal data. To accomplish this, we leverage the cross-entropy loss function.

$$Loss_{pos} = -\frac{1}{N} \sum_{i=1}^{N} \log(1 - p_i) \tag{11}$$

$$Loss_{neg} = -\frac{1}{N} \sum_{i=1}^{N} \log(p_i). \tag{12}$$

We employ binary cross-entropy to compute the Discriminator's loss. where $Loss_{pos}$ denotes the discrimination loss associated with normal data, while $Loss_{neg}$ represents the corresponding loss for abnormal data. The model's output is represented by $p_i$.

# TRAINING AND INFERENCE

## Model training

Algorithm 1 outlines the training procedure for our model. Initially, the Encoder processes the normal data and a randomly selected subset of abnormal data at time $t$ to derive their

latent space representations, denoted as $Z_{pos}$ and $Z_{neg}$ (line 5). Subsequently, based on these distributions, resampling techniques are employed to generate corresponding sample data (Line 6). These sampled data are then input into the Discriminator, and their respective discrimination results, $O^{ds}_{pos}$ and $O^{ds}_{neg}$, are obtained (Line 7). Concurrently, the sampled data from the latent space distribution of normal data is fed into the Decoder to reconstruct the original normal data, yielding the reconstruction output $O^{dc}_{pos}$ (Line 8). To evaluate the model's performance, we utilize Eq. (9) to compute the KL divergence ($Loss_{KL}$) between the normal distribution output by the Decoder and the predefined distribution, measuring their similarity (Line 9). Additionally, we employ Eq. (10) to calculate the reconstruction error ($Loss_{rec}$) of the normal data, assessing the accuracy of data reconstruction (Line 10). In terms of classification performance, we utilize Eq. (11) and Eq. (12) to calculate the binary cross-entropy losses ($Loss_{pos}$ and $Loss_{neg}$) for normal and anomalous data, respectively. These loss values reflect the model's performance in anomaly detection (Line 11). Finally, we aggregate these loss terms and update the model parameters iteratively using gradient descent algorithms (Lines 12, 13).

---

**Algorithm 1:** Training Process

**Require:** Encoder $E$, Decoder $DC$ and Discriminator $DS$, normal dataset $X_{pos}$ and anomaly dataset $X_{neg}$, iteration limit $n$.

1  Initialize weights $E, DC, DS$
2  $i \leftarrow 0$
3  **while** $i < n$ **do**
4      **for** $t=1$ to $T$ **do**
5          $Z_{pos}, Z_{neg} \leftarrow E(X^t_{pos}), E(X^s_{neg})$
6          $S_{pos}, S_{neg} \leftarrow Sample(Z_{pos}), Sample(Z_{neg})$
7          $O^{ds}_{pos}, O^{ds}_{neg} \leftarrow DS(S_{pos}), DS(S_{neg})$
8          $O^{dc}_{pos} \leftarrow DC(S_{pos})$
9          $Loss_{KL} = KL(Z_{pos}, Z_0)$
10          $Loss_{rec} = \left\| O^{dc}_{pos} - X^t_{pos} \right\|$
11          $Loss_{pos}, Loss_{neg} = BCELoss(O^{ds}_{pos}, \vec{0}), BCELoss(O^{ds}_{neg}, \vec{1})$
12          $Loss = Loss_{KL} + Loss_{Rec} + Loss_{pos} + Loss_{neg}$
13          Update weights of $E, DC, DS$ using Loss
14          $i \leftarrow i+1$
15      **end**
16  **end**

---

## Model inference

Using the trained anomaly detection model for inference (summarized in Algorithm 2), we define the anomaly score for a given data point within a specific time window of the

test set as follows:

$$Score = BCELoss(O^{ds}, \vec{0}) \tag{13}$$

As previously mentioned, the Decoder serves solely as an auxiliary component, aiding in the extraction of more comprehensive features from the Encoder. Consequently, during the inference phase, there is no need to feed the test data into the Decoder for reconstruction. Instead, our focus is on utilizing the Encoder to map the test data into the latent space effectively (line 2). Following this, we sample data from the latent space distribution and forward it to the Discriminator (line 3). The Discriminator then produces a set of vectors, where each value indicates the likelihood of the corresponding test data being anomalous (line 4). To derive the final anomaly score, we compute the cross-entropy loss between this vector and 0 (line 5). In alignment with established practices in prior research, we adopt the Peak Over Threshold (POT) method *Siffer et al. (2017)* for automatic threshold selection. The POT method is a statistical approach rooted in extreme value theory, which fits the data distribution to a generalized Pareto distribution. This enables us to determine an appropriate risk value that dynamically sets the threshold. Any anomaly score surpassing the POT threshold is deemed anomalous (line 6).

---

**Algorithm 2:** Inference Process

**Require:** Trained Encoder $E$, Decoder $DC$ and Discriminator $DS$, test dataset X.

1  **for** *t=1 to T* **do**
2     $Z \leftarrow E(X^t)$
3     $S \leftarrow Sample(Z)$
4     $O^{ds} \leftarrow DS(S)$
5     $Score = BCELoss(O^{ds}_{pos}, \vec{0})$
6     $y_t = 1(Score_t \geq POT)$
7  **end**

---

# EXPERIMENTS

## Datasets and metrics

- **Datasets**: To validate the efficacy of our proposed model, we employ three datasets: MSL, SMAP, and ETL. MSL and SMAP are publicly available datasets released by NASA (*O'Neill et al., 2010*) and have gained widespread utilization in anomaly detection tasks related to multivariate time series data. Additionally, the ETL dataset was collected through randomly and repeatedly executing 60 distinct ETL tasks *via* scripting in a commercial company's ETL system. The performance metrics dataset, which encompassed the utilization of critical hardware resources such as CPU, memory, and disk I/O, was gathered using the Prometheus tool. The overall duration of this dataset spans 104.5 h, with a collection frequency of once every five seconds. A detailed statistical overview of these three datasets is presented in Table 1.
- **Evaluation metrics**: We utilize precision, recall, and F1-Score as evaluation metrics to assess the model's performance. Previous works, such as those by *Tian, Su & Yin (2022)*,

**Table 1  Brief overview of datasets.**

| Datasets | Features | Train | Test | Anomalies |
|----------|----------|-------|------|-----------|
| MSL | 55 | 58,317 | 73,729 | 10.53% |
| SMAP | 25 | 135,183 | 427,617 | 12.79% |
| ETL | 12 | 54,361 | 20,881 | 18.62% |

*Zhao et al. (2020)*, and *Tuli, Casale & Jennings (2022)*, often adopt a lenient approach, termed "soft identification," where the entire abnormal interval is deemed correctly identified if only one point within it is marked as abnormal. However, this method can inflate the precision and recall scores, particularly when dealing with long abnormal intervals in the test set, leading to a potentially misleading performance evaluation. To address this limitation, we introduce a more stringent evaluation strategy known as "hard identification." Under this approach, the identification status of other points within an interval remains unaffected, even if the interval itself is marked as abnormal. This approach ensures a more accurate assessment of the model's true performance.

## Experimental setup
### Baselines
To highlight the performance of our proposed method, we compare it with multiple benchmark models, including both classic methods and currently popular deep learning algorithms, as follows:

- IF (*Liu, Ting & Zhou, 2008*): This technique employs an iTree binary search tree structure to isolate samples, subsequently leveraging these isolated sample points to detect anomalies.
- KNN (*Ramaswamy, Rastogi & Shim, 2000*): This method relies on the distance between a node and its k-th nearest neighbor to compute the anomaly score, offering an efficient approach to anomaly detection.
- MTAD-GAT (*Zhao et al., 2020*): Leveraging the power of GAT, this method learns temporal and spatial dependencies among variables. The anomaly score is derived from the combined reconstruction and prediction errors.
- TranAD (*Tuli, Casale & Jennings, 2022*): TranAD is an anomaly detection model rooted in the Transformer architecture, it utilizes both reconstruction and prediction errors as loss functions during training. During testing, the prediction error serves as the anomaly score.
- GDN (*Deng & Hooi, 2021*): This method uses embedding vectors to capture spatial relationships among variables, employing attention mechanisms for one-step prediction. The prediction error is considered the anomaly score.
- ImDiffusion (*Chen et al., 2023*): This innovative approach combines time series interpolation with diffusion models, enabling accurate and robust anomaly detection in multivariate time series data.

**Table 2   Some crucial hyperparameters of model training.**

| Hyperparameters | Value |
|---|---|
| Window size | 20 |
| Learning rate | 0.001 |
| Iterations | 50 |
| Alomaly proportion | 0.1 |

### Configuration

In this study, we implemented the proposed method using PyTorch 1.13.1 framework and CUDA 11.7 technology on an RTX A6000 GPU, leveraging the Adam optimizer for model training. Table 2 summarizes the crucial hyperparameters and their values for model training. For all datasets, we employed a unified set of experimental parameters. Specifically, the input window size was set to 20, as it was found to be an appropriate value for capturing temporal dependencies while avoiding excessive noise or computational complexity. A learning rate of 0.001 was chosen as it strikes a balance between stability and convergence speed. Our model achieved convergence after 50 training iterations across all three datasets. Furthermore, during training, we randomly selected 0.1 of the anomalous samples from the test set as the anomalous data for training. This proportion was chosen as we observed a significant improvement in the F1-Score when increasing the ratio up to 0.1, with diminishing gains thereafter.

In the specific implementation of the model, we configured the parameters for multiple key modules. Within the Encoder, we employed a one-dimensional convolutional kernel with a size of 7. For Spatial-feature Extractor, we utilized an optimized Graph Attention Network (GAT) that exhibited stronger feature representation capabilities (*Brody, Alon & Yahav, 2021*). Both the input and output dimensions of the GAT were set to be equal to the number of features in the dataset. The Temporal-feature Extractor consisted of a single layer LSTM network with an output dimension of 160. The Latent Distribution Mapper comprised two identical Multi-Layer Perceptrons (MLPs), each containing three layers. The input layer dimension matched the LSTM output dimension, the hidden layer dimension was set to 128, and the output layer dimension was 16. The activation function between the input layer and the hidden layer was the tanh function. Both the Decoder and the Discriminator were implemented using MLPs with three layers. The input layer dimension aligned with the output dimension of the Latent Distribution Mapper, the hidden layer dimension was 160, and the output layer dimension was equivalent to the number of features in the dataset.

## Experimental results

Tables 3 and 4 present the performance metrics of each model, utilizing the ''soft identification'' and ''hard identification'' methods, respectively. Given the varying threshold selection mechanisms among methods, this study meticulously tests each model's potential thresholds and reports the outcomes with the highest F1-Score. The tables present F1-Scores, emphasizing the peak value in bold and the second-highest in underline. In the soft identification approach, Table 3 reveals that EDD achieves superior performance

**Table 3  Performance of our models and benchmarks with soft identification.**

| Model | MSL | | | SMAP | | | ETL | | |
|---|---|---|---|---|---|---|---|---|---|
| | Pre | Rec | F1 | Pre | Rec | F1 | Pre | Rec | F1 |
| IF | 0.5574 | 0.9640 | 0.7064 | 0.3977 | 0.9108 | 0.5536 | 0.6453 | 0.8770 | 0.7435 |
| KNN | 0.4222 | 0.9653 | 0.5875 | 0.3117 | 0.9585 | 0.4704 | 0.5835 | 0.9423 | 0.7207 |
| MTAD-GAT | 0.9219 | 0.9660 | **0.9434** | 0.8362 | 0.9999 | 0.9107 | 0.786 | 0.8955 | 0.8372 |
| TranAD | 0.9142 | 0.7959 | 0.8509 | 0.7370 | 0.8430 | 0.7865 | 0.6703 | 0.7186 | 0.6936 |
| GDN | 0.9300 | 0.7826 | 0.8513 | 0.8687 | 0.5907 | 0.7032 | 0.7978 | 0.8953 | 0.8437 |
| ImDiffusion | 0.8844 | 0.8664 | 0.8753 | 0.8716 | 0.9662 | 0.9164 | 0.7852 | 0.8753 | 0.8237 |
| **EDD** | 0.9212 | 0.9168 | 0.9190 | 0.9180 | 0.9544 | **0.9358** | 0.9307 | 0.8819 | **0.9057** |

Notes.
Peak values are indicated in bold. Second-highest values are indicated with an underline.

**Table 4  Performance of our models and benchmarks with hard identification.**

| Model | MSL | | | SMAP | | | ETL | | |
|---|---|---|---|---|---|---|---|---|---|
| | Pre | Rec | F1 | Pre | Rec | F1 | Pre | Rec | F1 |
| IF | 0.1524 | 0.1446 | 0.1446 | 0.1532 | 0.2495 | 0.1898 | 0.3349 | 0.2427 | 0.2815 |
| KNN | 0.2052 | 0.3411 | 0.2562 | 0.1955 | 0.5145 | 0.2833 | 0.3045 | 0.2944 | 0.2994 |
| MTAD-GAT | 0.2166 | 0.4293 | 0.2880 | 0.1983 | 0.4487 | 0.2751 | 0.2129 | 0.8724 | 0.3423 |
| TranAD | 0.3688 | 0.0441 | 0.0789 | 0.1742 | 0.2896 | 0.2175 | 0.1973 | 0.3022 | 0.2387 |
| GDN | 0.1255 | 0.3445 | 0.1839 | 0.0948 | 0.1161 | 0.1044 | 0.2358 | 0.3515 | 0.2822 |
| ImDiffusion | 0.2095 | 0.4452 | 0.2849 | 0.1058 | 0.2358 | 0.1460 | 0.1866 | 0.2939 | 0.2283 |
| **EDD** | 0.5039 | 0.5144 | **0.5091** | 0.3149 | 0.4968 | **0.3855** | 0.6158 | 0.7487 | **0.6757** |

Notes.
Peak values are indicated in bold. Second-highest values are indicated with an underline.

on both the SMAP and ETL datasets, surpassing the second-best score by 3% and 7%, respectively. For the MSL dataset, it ranks second in F1-Score performance. Transitioning to the hard identification approach, as shown in Table 4, our proposed method surpasses all benchmark models across all datasets, with particularly impressive gains observed on the MSL and ETL datasets. Specifically, our method outperforms the second-ranked approach by 36% and 97% on the MSL and ETL datasets, respectively. A comparison of the two tables highlights the limitations of the evaluation approach used for previous models. Notably, several models that exhibit robust performance in the soft identification approach display subpar performance in the hard identification approach. This discrepancy is attributed to the soft identification evaluation approach, which conceals incorrect detections within a specific range if a model identifies an anomaly within that range. Conversely, the hard identification approach does not allow for such concealment, leading to a significant decline in the model's performance metrics.

## Ablation studies

To assess the effectiveness of each component in our EDD model, we conducted a series of ablation experiments. Specifically, we excluded the key configurations of each model component individually to observe the corresponding performance changes. To ensure

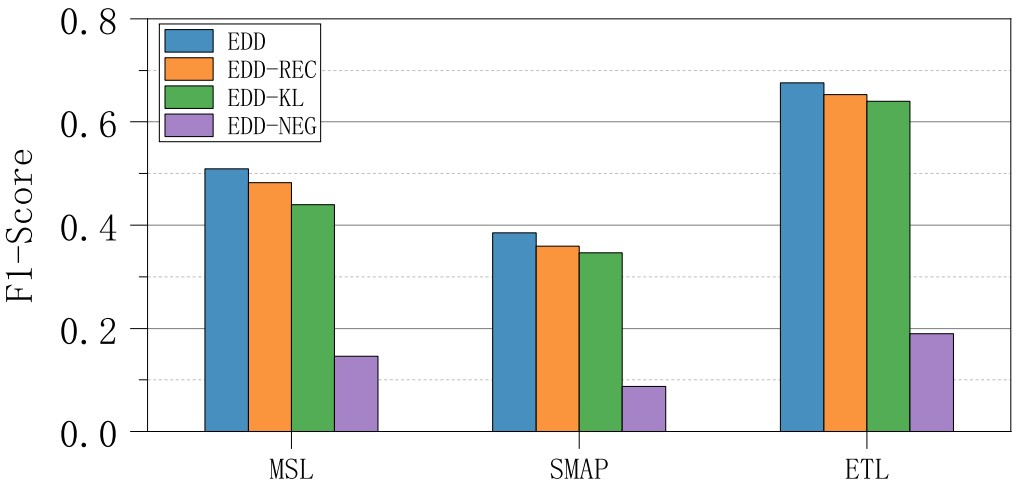

**Figure 2** Ablation results of ablation experiments.

rigorous evaluation, we utilized the F1-Score metric from the hard identification approach, which offers stricter criteria for performance assessment. This allowed us to precisely detect any performance degradation in the model.

In the first experiment, we disabled the reconstruction loss ($Loss_{rec}$) to explore the impact of the Encoder module on model performance. This variant was designated as EDD-REC. Subsequently, we turned off the KL loss ($Loss_{KL}$) to investigate whether clustering normal data effectively enhances model performance. This variant was labeled as EDD-KL. Lastly, we eliminated the negative loss ($Loss_{neg}$) to assess the guidance provided by abnormal data on the model. This variant was denoted as EDD-NEG.

The experimental results are presented in Fig. 2. These findings indicate that each key module in our model contributes positively to improving overall performance. Notably, the introduction of a limited amount of abnormal data had the most significant impact on enhancing model performance. This underscores the critical role of abnormal data in guiding the model to identify anomalous patterns effectively.

## Data distinguishability

In anomaly detection, algorithms typically aim to establish a threshold that effectively separates abnormal data from normal data, solely relying on anomaly scores. This approach disregards the distinctiveness between normal and abnormal data. However, a model that demonstrates superior discrimination between these two categories is likely to excel in tasks that demand greater distinctiveness, thereby indicating its overall superior performance. To compare our approach with others, we conducted studies using the highly effective deep learning method, MTAD-GAT.

To visualize the anomaly scores on the test set, we generated box plots, as shown in Fig. 3. Specifically, subfigure A, B, and C represent the box plots for the MSL, SMAP, and ETL datasets respectively. The figures clearly illustrate that the proposed EDD method

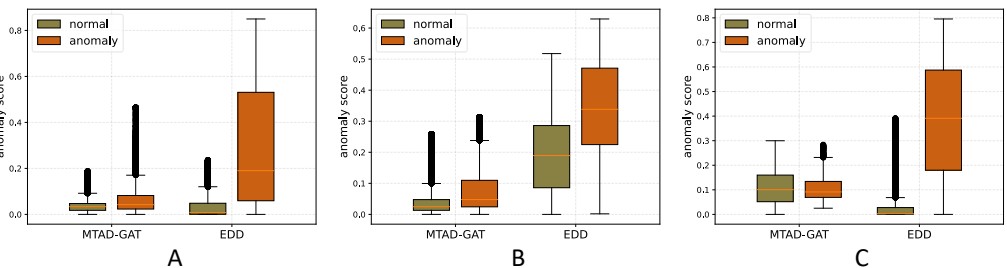

**Figure 3** Box line plot of MTAD and EDD's anomaly scores for normal and abnormal data.

exhibits a more pronounced distinction between normal and abnormal data compared to MTAD-GAT. This enhanced distinguishability is crucial for accurate anomaly detection.

Additionally, while analyzing the ETL dataset (Fig. 3C), we observed that MTAD-GAT did not exhibit clear distinguishability between normal and abnormal data. Despite this, the F1-Score, computed using the soft identification evaluation method, surprisingly reached a remarkably high score of 0.8372. This apparent paradox further validates the concern raised in this article: that the soft identification evaluation approach may not always provide an accurate assessment of model performance. Therefore, this article introduces a hard identification evaluation approach to ensure a more precise performance evaluation of the model.

## Latent space distribution

The idea of implementing anomaly detection in this article is to distinguish normal data from abnormal data based on their different probability distributions in the latent space. In the Encoder of the EDD model, we use the $Loss_{KL}$ to map the distribution of normal data in the latent space to a normal distribution, while making no constraints on the distribution of abnormal data. This allows the normal data to be more clustered in the latent space, and the abnormal data to be randomly distributed. The Discriminator only needs to learn the difference between normal data and a small amount of abnormal data to distinguish most unseen anomalies. We visualized the encoding in the latent space on a 2D plane, as shown in Fig. 4. The visualization results are consistent with our expectations. The normal data (green points) are clustered at one point, and the model's seen abnormal data (yellow crosses) and unseen abnormal data (purple rectangles) are randomly distributed in the latent space.

## Anomaly proportion

The core of the algorithm in this article lies in optimizing the model's ability to detect anomalies through a small amount of anomalous data. However, in real-world applications, the scarcity and difficulty of acquiring anomalous labels have become crucial factors restricting the improvement of model performance. To deeply explore the specific impact of the proportion of anomalous data on model performance, we designed an anomalous data ratio parameter $\gamma$ in the experiment, which adjusts the proportion of anomalous data extracted from the test set for training to the total amount of anomalous data.

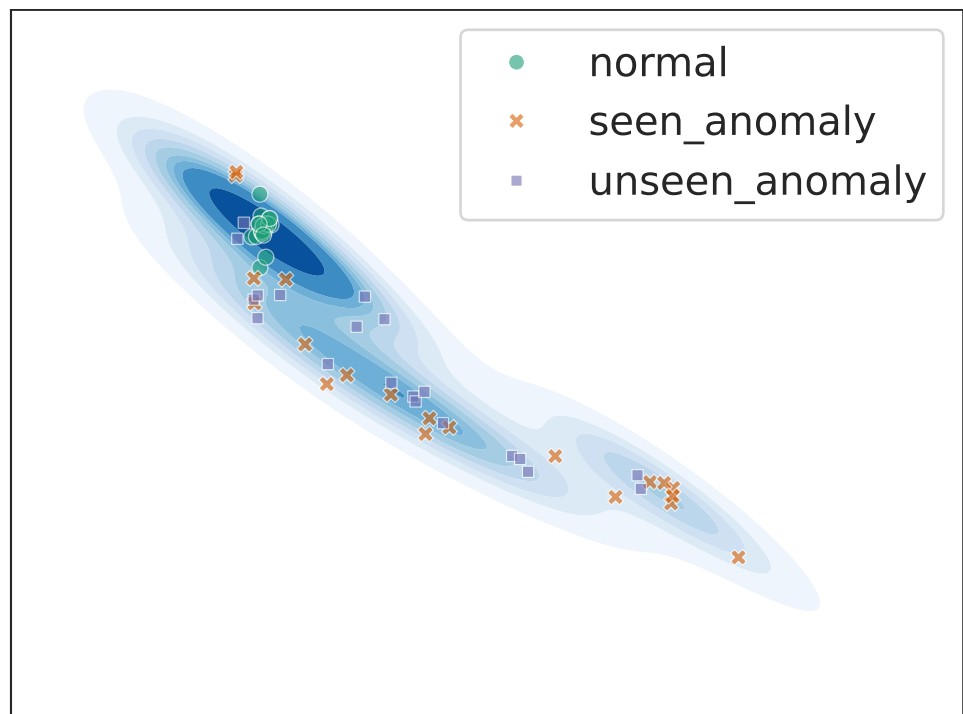

**Figure 4** **Probability distribution of data in the latent space.**

The experimental results, as shown in Fig. 5, clearly demonstrate the model's F1-Score performance under different anomalous data ratios.

The figure shows that in the absence of anomalous data for training, the model's performance on numerous datasets is poor, emphasizing the importance of anomalous data in model training. However, by incorporating a modest bit of aberrant data as training guidance, the model's performance improves dramatically. Remarkably, with only 1% of anomalous data included, the model's F1-Score increased by more than three times on average. This conclusion emphasizes the relevance of crucial information contained in anomalous data for improving the model's anomaly detection capacity, as well as providing useful insights into how to successfully employ limited anomalous data in actual applications. Furthermore, the experimental results also show that even with a very low anomalous data ratio (around 5%), the model can still effectively improve its anomaly recognition performance. This finding further validates the feasibility of deploying the algorithm in practical applications.

## CASE STUDY

In this section, we delve into two representative cases to illustrate the strengths and limitations of the EDD model in detecting anomalies in multivariate time series data.

Firstly, Fig. 6 showcases a successful anomaly detection case by the EDD model. Within the highlighted red segment, while individual metrics appeared normal, the model's ability

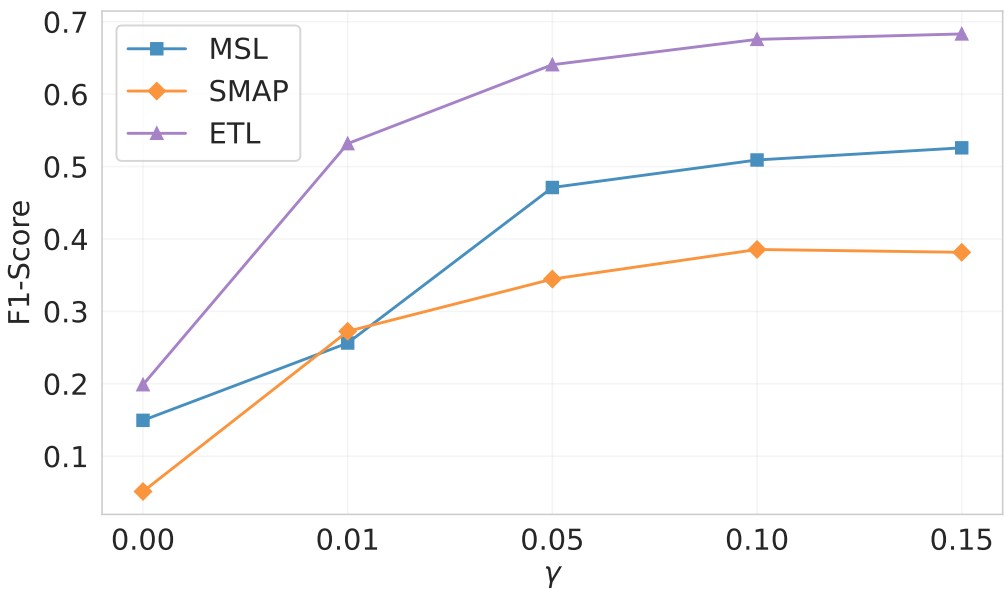

**Figure 5  Impact of the proportion of anomalous data on model performance.**

to capture complex relationships among features enabled it to identify a potential anomaly. Specifically, despite a surge in the number of ETL tasks being executed on a node, the CPU usage remained abnormally low. Further analysis revealed concurrent increases in dirty memory data and the number of IO operation components, suggesting a possible IO bottleneck. This case underscores the EDD model's superiority in considering intricate relationships among multiple variables and provides valuable insights for subsequent root cause analysis.

However, Fig. 7 presents a case where the EDD model falsely classified normal data as anomalous. During the red segment, all metrics exhibited significant spikes, prompting the model to flag it as anomalous. Nevertheless, a closer look at the tasks being executed on the node revealed that these spikes were actually caused by a sudden influx of tasks, a normal occurrence rather than an anomaly. This infrequent event in historical data confounded the model, leading to a misclassification. This case highlights the limitations of the EDD model in handling rare events and suggests that future research should incorporate more historical data and domain knowledge to enhance the model's accuracy and robustness.

In summary, through the analysis of these two cases, we have gained a deeper understanding of the EDD model's strengths and weaknesses in detecting anomalies in multivariate time series data. Future research will focus on further refining the model to improve its performance in practical applications.

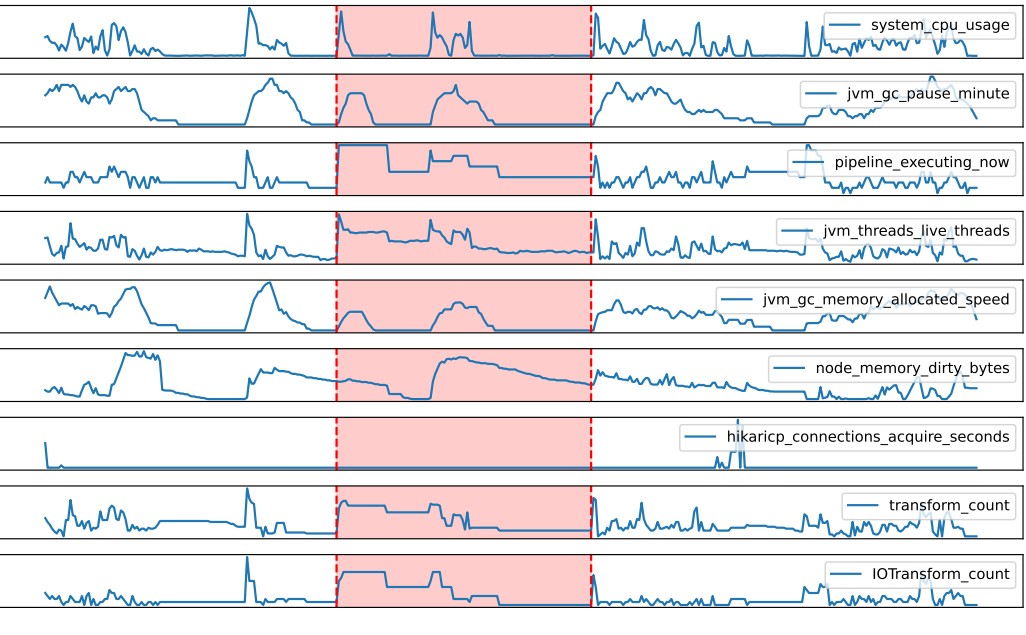

**Figure 6** True positive case in anomaly detection with the EDD model.

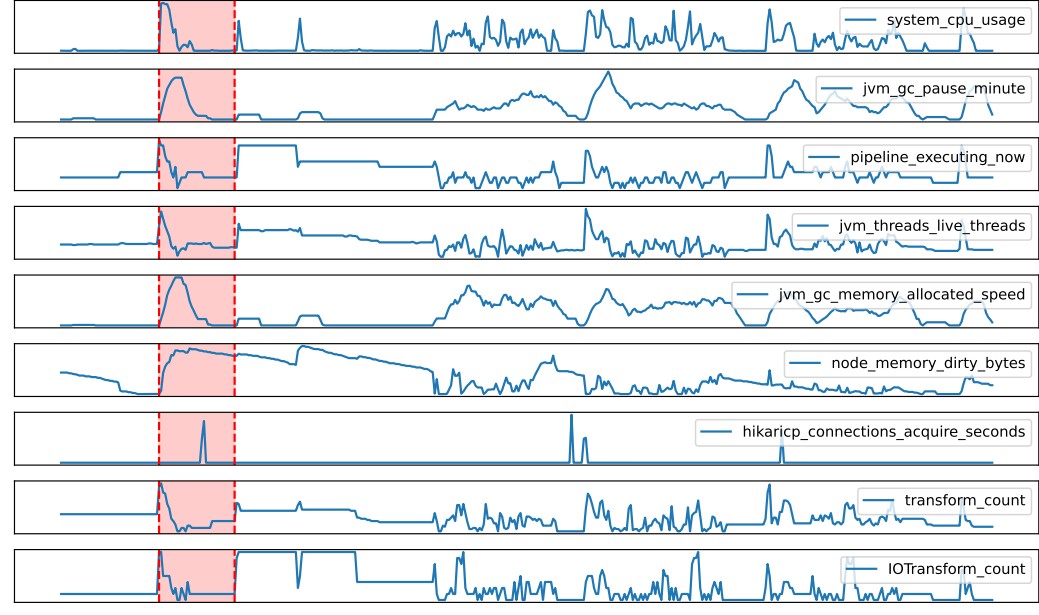

**Figure 7** False positive case in anomaly detection with the EDD model.

## CONCLUSION AND FUTURE WORKS

In this article, we propose a novel multivariate time series anomaly detection model named EDD that incorporates anomaly perception capabilities. By leveraging a graph attention

network and LSTM, our EDD model effectively extracts spatial and temporal features from sequences, respectively. A key advantage of our approach is the utilization of a limited amount of easily obtainable abnormal data, which guides the model to enhance its discrimination between normal and abnormal time series. Through extensive evaluation on multiple datasets, our method consistently outperforms other representative approaches in most scenarios. Moreover, our model exhibits superior separability between normal and abnormal data, indicating its superior accuracy in distinguishing between the two. We further validate the effectiveness of each key module through ablation experiments, confirming their contributions to the overall performance.

Our work has built a preliminary foundation for the efficient utilization of limited labeled data for anomaly detection, and further work can be carried out in the following directions in the future: Firstly, enhancing the utilization of the available labeled data can lead to improved detection accuracy. Current methods may not fully capitalize on the information contained in these limited samples. Future research can investigate techniques to effectively leverage the labeled data, such as semi-supervised learning methods or data augmentation strategies tailored for anomaly detection. Secondly, exploring more advanced feature extraction techniques can further improve the model's ability to distinguish anomalies. Current methods rely on graph attention networks and LSTMs to capture spatial and temporal features, but there may be other techniques or combinations of techniques that can better capture the underlying patterns in the data. Future work can investigate novel feature extraction methods or combinations of existing techniques to improve detection performance. Thirdly, addressing the challenges posed by the randomness of abnormal data distribution remains a key area for future research. While this work provides a foundation for understanding these challenges, there is still room for improvement in controlling the distribution of abnormal data in the latent space. Future algorithms can aim to develop more robust and consistent mechanisms for encoding anomalies, ensuring that they are consistently separated from normal data, regardless of their specific distribution patterns. Overall, the directions outlined above represent promising areas for future research in anomaly detection, building upon the foundation established by this work.

### Funding
This work was funded by the National Natural Science Foundation of China (No. 61562010 and 71964009), the Research Platforms and Projects of Universities in Guangdong Province - Youth Innovative Talents (No. 2022KQNCX071), and the Research Projects of the Science and Technology Plan of Guizhou Province (No. [2021]449,[2021]261, [2023]010, [2023]276, and [2023]338). The funders had no role in study design, data collection and analysis, decision to publish, or preparation of the manuscript.

### Grant Disclosures
The following grant information was disclosed by the authors:
The National Natural Science Foundation of China: 61562010, 71964009.

The Research Platforms and Projects of Universities in Guangdong Province - Youth Innovative Talents: No. 2022KQNCX071.
The Research Projects of the Science and Technology Plan of Guizhou Province: [2021]449, [2021]261, [2023]010, [2023]276, [2023]338.

## Competing Interests

The authors declare there are no competing interests.

## Author Contributions

- Dong Wei conceived and designed the experiments, performed the experiments, analyzed the data, performed the computation work, prepared figures and/or tables, authored or reviewed drafts of the article, and approved the final draft.
- Wu Sun performed the experiments, prepared figures and/or tables, and approved the final draft.
- Xiaofeng Zou performed the experiments, performed the computation work, prepared figures and/or tables, and approved the final draft.
- Dan Ma analyzed the data, prepared figures and/or tables, and approved the final draft.
- Huarong Xu analyzed the data, prepared figures and/or tables, and approved the final draft.
- Panfeng Chen analyzed the data, prepared figures and/or tables, and approved the final draft.
- Chaoshu Yang conceived and designed the experiments, authored or reviewed drafts of the article, and approved the final draft.
- Mei Chen conceived and designed the experiments, authored or reviewed drafts of the article, and approved the final draft.
- Hui Li conceived and designed the experiments, performed the computation work, authored or reviewed drafts of the article, and approved the final draft.

## Data Availability

The raw data and code are available in the Supplemental Files.

## Supplemental Information

Supplemental information for this article can be found online at http://dx.doi.org/10.7717/peerj-cs.2172#supplemental-information.

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
