# Peer review of "An anomaly detection model for multivariate time series with anomaly perception"

_PeerJ Computer Science, doi:10.7717/peerj-cs.2172_

## Round 0.1 · original submission · Major Revisions

I have received reviews of your manuscript from scholars who are experts on the cited topic. They find the topic interesting; however, several concerns must be addressed regarding discussions of the proposed methodology, related works, experimental results, and the English language. These issues require a major revision. Please refer to the reviewers’ comments at the end of this letter; you will see that they advise you to revise your manuscript. If you are prepared to undertake the work required, I would be pleased to reconsider my decision. Please submit a list of changes or a rebuttal against each point that is being raised when you submit your revised manuscript.

Thank you for considering PeerJ Computer Science for the publication of your research.

With kind regards,

**Language Note:** The Academic Editor has identified that the English language must be improved. PeerJ can provide language editing services - please contact us at [email protected] for pricing (be sure to provide your manuscript number and title). Alternatively, you should make your own arrangements to improve the language quality and provide details in your response letter. – PeerJ Staff

Reviewer 1 ·

Basic reporting

Overall, the paper presents the EDD model, an innovative approach for multivariate time series anomaly detection. The methodology is robust and experimental validation supports the superior performance of the proposed model. However, the paper would benefit from minor improvements in the clarity of the language and a clearer discussion of the limitations.

Experimental design

The proposed EDD model integrates a graph attention network with LSTM, which allows the extraction of spatial and temporal features from multivariate time series data.

Validity of the findings

The methodology is described in sufficient detail to allow replication of the study.
The experimental validation is comprehensive, performed on three different datasets, and the results demonstrate the superior performance of the EDD model in multivariate time series anomaly detection.

Additional comments

1. In order to develop the article, it is necessary to first work on language and expression. It is important to correct errors in the English language and review sentence structures to make the text more understandable.
2. Language can be improved for clarity. For example, lines 23, 77, 121, and 128 could benefit from better wording. The existing expressions make it difficult to understand.
3. Limitations of the proposed model are not explicitly discussed. It would be helpful to acknowledge potential disadvantages or areas for improvement.
4. While the methodology section provides sufficient detail, additional information regarding the training process and hyperparameter selection may increase the reproducibility of the study.
5. Discussion: is one of the most important parts of the article. There is a lack of discussion about the practical implications and potential applications of the EDD model. Including a section on real-world case studies or scenarios to which the model can be applied will increase the relevance and applicability of the research. It would benefit from a more critical discussion of the limitations of the proposed EDD model. Addressing potential weaknesses and making recommendations for future research will increase the overall impact of the study.

Cite this review as

Reviewer 2 ·

Basic reporting

The manuscript is written in clear and professional English. It provides sufficient background and context about the EDD model and its application in multivariate time series anomaly detection. The structure of the article is professional, with well-organized sections, figures, and tables. The paper is self-contained, presenting relevant results to the hypothesis. The results are formally presented with clear definitions of all terms and theorems, and detailed proofs. However, there are a few areas that need to be improved to increase the flow and fluency of the writing.

Experimental design

The paper presents original primary research within the Aims and Scope of the journal. The research question is well defined, relevant, and meaningful. It focuses on the effectiveness of the EDD model in leveraging limited anomaly samples to improve the performance of multivariate time series anomaly detection. The study fills an identified knowledge gap by introducing a novel deep learning model that integrates a graph attention network with LSTM to extract spatial and temporal features from multivariate time series data. The investigation is rigorous, performed to a high technical and ethical standard. The methods, including the architecture of the EDD model and the experimental validation process, are described with sufficient detail and information to replicate.

Validity of the findings

The underlying data appears to be robust, statistically sound, and controlled. The conclusions are well stated, linked to the original research question, and limited to supporting results. The study concludes by acknowledging the robust performance of the EDD model and highlights the implications of the soft identification and hard identification evaluation approaches on model performance. However, there is a discrepancy in the assumption about the distribution of normal data points. The author assumes normal data will be Gaussian distribution around some center points, but equation 9 defines the loss function as a KL divergence between a data distribution and a pre-defined Gaussian distribution. The author needs to clarify whether they are assuming the normal data points follow a Gaussian distribution or a mixture of Gaussian distributions.

Additional comments

The manuscript presents a comprehensive overview of the EDD model’s architecture, experimental validation, and ablation studies, showcasing its effectiveness in multivariate time series anomaly detection. However, the “Related works” section is a bit too brief and does not mention the baseline models used for benchmarking. The author should explain the motivation behind proposing a new work over the existing methods. The visualizations of the data box-plot and latent space distribution in the experiment result discussion section are helpful for understanding the datasets and the model’s effectiveness. The manuscript provides valuable insights into the EDD model’s performance characteristics and potential future directions for improvement.

Cite this review as

---

## Round 0.2 · accepted · Accept

I am pleased to inform you that your work has now been accepted for publication in PeerJ Computer Science.

Please be advised that you are not permitted to add or remove authors or references post-acceptance, regardless of the reviewers' request(s).

Thank you for submitting your work to this journal. On behalf of the Editors of PeerJ Computer Science, we look forward to your continued contributions to the Journal.

With kind regards,

Reviewer 1 ·

Basic reporting

The V1 , which is fixed after feedback, looks much better. Thanks for fixing the review comments.

Experimental design

The updated document offers an expanded explanation for the selection of the alternative.

Validity of the findings

The revised version has provided more background on choosing the replacement.

Additional comments

Thanks for the opportunity to review the revised manuscript. Based on the revised comments as well as data points I recommend this paper for publication.

Cite this review as

Reviewer 2 ·

Basic reporting

The manuscript is well-written in clear and professional English, with a professional structure, figures, and tables.

Experimental design

The research question is well-defined, relevant, and meaningful, and the investigation is rigorous and well-executed.

Validity of the findings

The underlying data appears robust, statistically sound, and controlled, and the conclusions are well-stated and linked to the original research question.

Additional comments

I am pleased to see that the authors have thoroughly addressed all the issues raised in my previous review. The manuscript is now clear, well-organized, and easy to follow. The experimental design, including the research question, methods, and validation process, are well-executed and rigorous. The authors have successfully clarified the assumption about the distribution of normal data points, and the "Related Works" section has been strengthened to provide a more comprehensive understanding of the research background and motivation. Overall, I recommend acceptance of the manuscript.

Cite this review as